# Plasma Beam Dumps for the EuPRAXIA Facility

**Guoxing Xia [1,2,\*], Alexandre Bonatto [1,2,\*], Roger Pizzato Nunes [3], Linbo Liang [1,2], Oscar Jakobsson [1], Yuan Zhao [1,2], Barney Williamson [1,2], Can Davut [1,2] and Xueying Wang [1]**

[1] Department of Physics and Astronomy, University of Manchester, Manchester M13 9PL, UK; linbo.liang@postgrad.manchester.ac.uk (L.L.); oscar.jakobsson.academic@gmail.com (O.J.); yuan.zhao@manchester.ac.uk (Y.Z.); barney.williamson@postgrad.manchester.ac.uk (B.W.); can.davut@manchester.ac.uk (C.D.); xueying.wang-6@student.manchester.ac.uk (X.W.)

[2] Cockcroft Institute, Warrington WA4 4AD, UK

[3] Departamento de Engenharia Elétrica, Escola de Engenharia, Universidade Federal do Rio, Grande do Sul, Porto Alegre 90035-190, Brazil; roger.pizzato@ufrgs.br

\* Correspondence: guoxing.xia@manchester.ac.uk (G.X.); Alexandre.Bonatto@manchester.ac.uk (A.B.)

**Abstract:** Beam dumps are indispensable components for particle accelerator facilities to absorb or dispose beam kinetic energy in a safe way. However, the design of beam dumps based on conventional technology, i.e., energy deposition via beam–dense matter interaction, makes the beam dump facility complicated and large in size, partly due to the high beam intensities and energies achieved. In addition, specific methods are needed to address the radioactive hazards that these high-power beams generate. On the other hand, the European Plasma Research Accelerator with eXcellence in Application (EuPRAXIA) project can advance the laser–plasma accelerator significantly by achieving a 1–5 GeV high-quality electron beam in a compact layout. Nevertheless, beam dumps based on the conventional technique will still produce radiation hazards and make the overall footprint less compact. Here, a plasma beam dump will be implemented to absorb the kinetic energy from the EuPRAXIA beam. In doing so, the overall compactness of the EuPRAXIA layout could be further improved, and the radioactivity generated by the facility can be mitigated. In this paper, results from particle-in-cell simulations are presented for plasma beam dumps based on EuPRAXIA beam parameters.

**Keywords:** beam dump; laser plasma accelerator; plasma beam dump

## 1. Introduction

The European Plasma Research Accelerator with eXcellence in Application (EuPRAXIA) is an EU design study proposed with the aim to produce a conceptual design for a worldwide first 1–5 GeV plasma-based accelerator with an industrial-level beam quality and user areas [1]. One of the important advantages of this project is the compactness of the facility. As will be discussed in this work, with a plasma beam dump, the overall facility footprint can be reduced further, and the radioactive hazards can be diminished significantly.

The development of compact, high-quality electron accelerators based on laser–plasma technology has already attracted great interest worldwide since the initial idea was proposed by Tajima and Dawson more than 40 years ago [2–6]. The basic principle behind this is to utilize the strong electric field associated with collective electron oscillations in the plasma to accelerate either an internally or an externally injected electron beam behind the driver pulse. Due to the collective nature of this technique, it is possible to achieve an extremely high acceleration gradient, usually more than 3-fold higher than the RF field used in conventional accelerators. Nowadays, electron beams of several GeV

can routinely be achieved in laser wakefield accelerators (LWFA) within centimetre-long plasmas by using terawatt ($10^{12}$ Watt) or petawatt ($10^{15}$ Watt) laser drivers [7–9].

On the other hand, the use of plasma wakefields for deceleration of relativistic beams has not been fully explored ever since. In 2010, Tajima et al. proposed the collective deceleration of beams in plasmas for the first time [10]. The idea is to utilize large decelerating wakefields, with amplitudes as high as those of accelerating fields, to absorb beam energy as fast as possible. This would allow beam deceleration to be achieved in a short distance compared to equivalent conventional beam dumps. Moreover, this could mitigate conventional beam dump requirements, which usually suffer from complicated design and large sizes (and costs) when the beam power is high. In addition, the use of low-density plasma greatly reduces radio activation hazards compared to conventional beam dumps, in which energetic particles interact with dense media such as metals, graphite or water, causing nuclear reactions, the production of secondary particles, and generating radiation.

This work is organised as follows. In Section 2, there is a brief discussion about plasma beam dumps. In Section 3, simulation results for passive plasma beam dumps designed for the 1 and 5 GeV EuPRAXIA beams are presented and discussed. In Section 4, final discussions and conclusions are detailed.

## 2. Plasma Beam Dumps

Generally, there are two types of plasma beam dumps—the so-called passive plasma beam dump (PPBD), and the active plasma beam dump (APBD) [11,12]. For the PPBD, a relativistic particle bunch propagates in an undisturbed plasma and excites its own wakefield. Consequently, the head of the bunch will experience no decelerating field due to the finite response time of the plasma, while particles at the bunch tail will experience a decelerating field. After some time, the fraction of the bunch experiencing the maximum decelerating field will become non-relativistic, and it will fall behind the rest of the bunch until it reaches an accelerating phase of the wakefield. This causes beam re-acceleration, which eventually leads to saturation of beam net energy loss [9,13,14]. In order to eliminate beam re-acceleration, several schemes have been proposed, which include inserting foils in the plasma to absorb the re-accelerated particles, and tailoring the plasma density along the beam propagation direction to change the relative phases of wakefield along the beam driver. Recent studies have shown that the beam energy deposition in plasma can be greatly enhanced through finely tailoring the plasma densities [13,14]. On the other hand, in the APBD this beam re-acceleration is eliminated. In this scheme, a laser pulse is employed to excite a wakefield in the plasma prior to beam propagation, in such a way that the combination of both laser-driven and beam-driven wakefields flattens the decelerating field along the bunch. This enables a quasi-uniform energy extraction, thus preventing the formation of re-acceleration peaks [11,12]. Although energy extraction is more efficient in the APBD, the need for a laser pulse and precise synchronization between the laser and the beam makes this scheme far more complex to implement experimentally than the PPBD.

## 3. Simulation Results of Plasma Beam Dumps for EuPRAXIA Beams

In order to simplify the design and implementation of a plasma beam dump for the EuPRAXIA facility, we propose the adoption of a passive scheme. We aim to absorb most of the energy from the electron bunch by tailoring the plasma density profile. Typical EuPRAXIA beam parameters [1,15] used in our studies are listed in Table 1. Here, two sets of beam parameters are considered in our simulation—one with a beam energy of 1 GeV and the other of 5 GeV. Other beam parameters are the same. This corresponds to a beam density of ~$3.0 \times 10^{18}$ cm$^{-3}$. The Fourier–Bessel particle-in-cell (FBPIC) code [16] is used to perform simulations of beam–plasma interaction. Although particles in FBPIC have 3D cartesian coordinates, this code adopts a spectral solver, which uses a set of 2D radial grids, each of them representing an azimuthal mode, $m$ ($m = 0, 1, \ldots$). The first mode, $m = 0$, represents axisymmetric fields, i.e., fields with cylindrical symmetry, with no dependence on the azimuthal angle, $\theta$. Additional modes can be used to represent departures from the cylindrical symmetry (for example,

a linearly polarised laser can be represented by the second azimuthal mode, $m = 1$). Among the many interesting features implemented in FBPIC, it is worth highlighting the mitigation of spurious numerical dispersion by the spectral solver algorithm, including the zero-order numerical Cherenkov effect [17], and the adoption of the openPMD meta data standard [18]. The simulations presented in this document were performed using the first azimuthal mode ($m = 0$). Hence, a 2D axisymmetric geometry is adopted. The longitudinal simulation domain is $-2.5\,\pi/k_p \leq \xi \leq 5\,\sigma_\xi$, where $\xi \equiv z - ct$ is the co-moving coordinate and $\sigma_\xi$ is the longitudinal bunch root mean square (RMS) size. Transversally, the domain is $-20\,\sigma_r \leq x$, $y \leq 20\,\sigma_r$, where $x$ and $y$ are the transverse coordinates, and $\sigma_r$ is the transverse RMS bunch length. This domain is enough to evaluate the relevant dynamics, such as the formation of re-acceleration peaks and the behaviour of defocused particles. The longitudinal and transverse resolutions are $\sigma_\xi/20$ and $\sigma_r/20$, respectively, and the total number of particles per cell is 4, being 2 along each coordinate $z$ and $r$.

**Table 1.** EuPRAXIA beam parameters used in simulation.

| Beam Energy | 1 GeV | 5 GeV |
|---|---|---|
| Bunch charge | 30 pC | 30 pC |
| Transverse bunch size ($\sigma_r$) | 1.4 μm | 1.4 μm |
| Longitudinal bunch length ($\sigma_\xi$) | 2.0 μm | 2.0 μm |
| Energy spread | 1.0% | 1.0% |
| Angular divergence (rad) | $1.0 \times 10^{-5}$ | $1.0 \times 10^{-5}$ |

*3.1. Plasma Beam Dumps for the 1 GeV Beam*

As a first step, we choose a plasma density of $n_0 = 9.9 \times 10^{17}$ cm$^{-3}$. For this plasma density and considering the beam charge and dimensions presented in Table 1, a beam–plasma density ratio of $n_b/n_0 \simeq 3.1$ is obtained. The magnitude of this ratio defines the regime of the wakefield excited by the beam propagating in the plasma. If $n_b/n_0 \ll 1$, the wakefield presents a perfectly harmonic aspect, and is said to be in the linear regime. As this ratio increases, the wakefield changes from a sinusoidal to a sawtooth-like shape. Although no longer harmonic, under certain conditions, some analytical estimates for the wakefield still hold for $1 \lesssim n_b/n_0 \lesssim 10$ [19]. Therefore, the propagation regime within the aforementioned interval of $n_b/n_0$ can be classified as ranging from the quasi-linear, or quasi-non-linear [20], to the non-linear regime. This is the case for the parameters chosen for this work. Besides the wakefield propagation regime, the choice of plasma density must also take into account the electron bunch size. While the longitudinal wakefield oscillates from decelerating (positive electric field) to accelerating (negative electric field) phases, the transverse wakefield alternates from focusing to defocusing phases. For a plasma beam dump, a density is chosen that can contain the whole bunch within the first wakefield phase, which is longitudinally decelerating and transversely focusing. This ensures that the bunch will be simultaneously decelerated and focused as it propagates in the plasma. With all the parameters defined, a particle-in-cell (PIC) simulation is performed.

Simulation results show that the particles in the bunch lose their energies quickly. After propagating approximately 6 cm in the plasma, the particles at the tail of the bunch lose most of their energies. This corresponds to an average decelerating gradient of approximately 16.7 GeV/m. The decelerated particles suffer phase slippage towards the next accelerating phase of the longitudinal wakefield. As a result, the bunch length increases during energy dumping. If the bunch propagates further in the plasma, the decelerated particles will reach an accelerating phase of the wakefield, gaining energy from it. This energy gain at the tail of the bunch compensates the energy loss from other bunch regions, saturating the net beam energy extraction. The distance at which this saturation starts (6 cm, in our case) is defined as the saturation distance. Figure 1 shows the beam longitudinal phase space after propagating 6 cm in plasma, in panel (a), and the beam energy spectrum, in panel (b). It can be seen in panel (a) that, while the beam energy at the head of the bunch ($\xi > 0$ μm) does not change, the particles at the tail ($\xi \simeq -20$ μm) start gaining energy. Figure 1a also shows a secondary spike of

higher-energy particles, located within the region $-10\ \mu\text{m} \lesssim \xi \lesssim -5\ \mu\text{m}$, with energies ranging from zero to approximately 0.6 GeV. This is due to the fact that, in this region, the wakefield is rapidly decreasing towards zero, and hence providing weaker decelerating gradients compared to particles in other regions. Figure 1b also provides quantitative information regarding the beam charge and energy distribution, hence complementing the beam phase space shown in Figure 1a. From Figure 1b, it can be seen that a large fraction of beam particles is decelerated to low energies, while a certain amount remains distributed up to the initial beam energy.

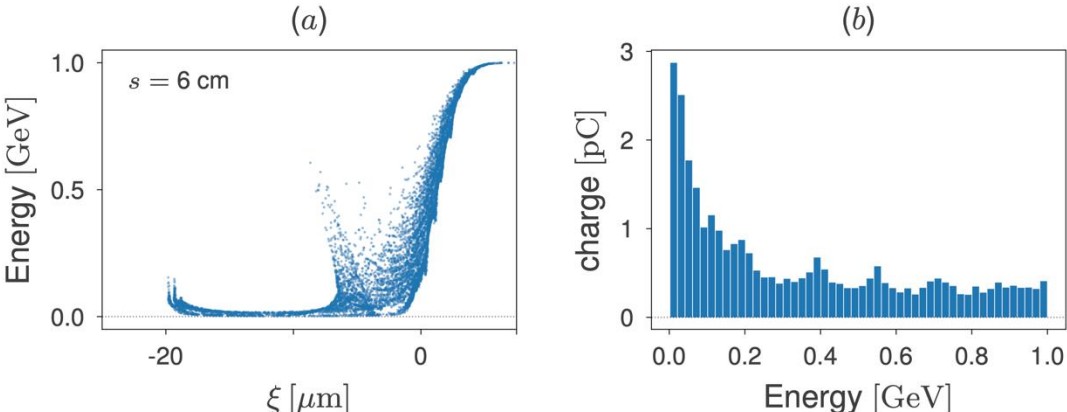

**Figure 1.** (**a**) The EuPRAXIA 1 GeV beam longitudinal phase space and (**b**) energy spectrum, both plotted after 6 cm propagation in plasma.

In order to avoid energy loss saturation, the particles in the bunch gaining energy from the wakefield accelerating phase can be eliminated by tailoring the plasma density profile. By continuously increasing the plasma density, the plasma wavelength is continuously shortened. If observed from the co-moving coordinate $\xi$, it looks like the wakefield phases are moving towards the head of the bunch as the plasma wavelength is reduced. Then, a defocusing phase, "moving" towards the bunch head in the co-moving coordinate $\xi$, can transversely eject particles previously located in an accelerating wakefield phase. Consequently, re-acceleration peaks are eliminated [14].

Figure 2 shows a typical plasma density profile, in which the density increases in a non-linear fashion from $n_0 = 9.9 \times 10^{17}\ \text{cm}^{-3}$, at 6 cm, to $10\ n_0 = 9.9 \times 10^{18}\ \text{cm}^{-3}$, at approximately 17 cm. This particular plasma density profile can be obtained by imposing a constant rate of change for the plasma wavelength $\lambda$ with respect to the propagation distance $s$, i.e., $d\lambda/ds = \text{constant}$ [14].

Figure 3a shows the beam longitudinal phase space after 16 cm propagation in plasma. It is found that the re-acceleration peak, as shown in Figure 1a, is eliminated, and particles continue to lose their energies in the plasma. Moreover, particles previously observed in the region of the beam phase space in which beam particles were less affected by the decelerating wakefield ($-10\ \mu\text{m} \lesssim \xi \lesssim 0\ \mu\text{m}$, Figure 1, panel a) are also impacted by the plasma density tuning. Figure 3a shows that this region is now smaller, approximately $\xi = 0\ \mu\text{m}$, also with overall smaller energies. The energy spectrum of the particles which remain inside the simulation domain (7 pC, at this propagation distance) is represented by the blue histogram of Figure 3b. These are the same particles shown in the beam longitudinal phase space plotted in Figure 3a. The prominent peak that can be seen at approximately 0.1 GeV is caused by the cumulative effect of particles being continuously decelerated due to the tailored plasma density profile. The light-red shaded region of Figure 3b shows the energy spectrum for the whole beam, including the transversely ejected particles.

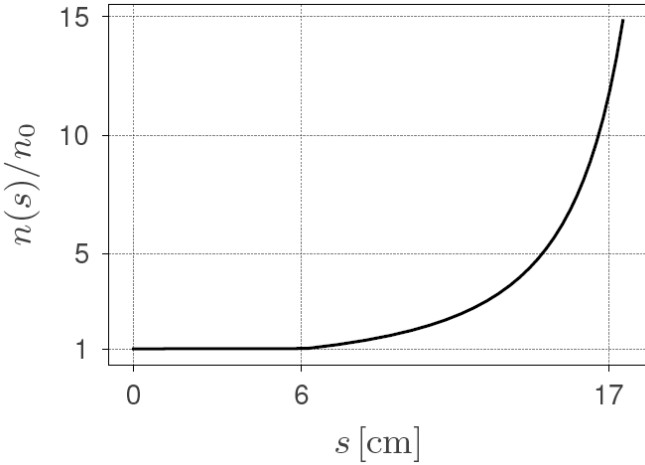

**Figure 2.** A tailored plasma density profile designed to eliminate the re-acceleration of particles in the bunch tail in the EuPRAXIA 1 GeV beam. The plasma density exhibits uniform behaviour until $s = 6$ cm, followed by a non-linear growth that reaches a 10 times higher density at $s = 16.8$ cm compared to the former uniform plasma density.

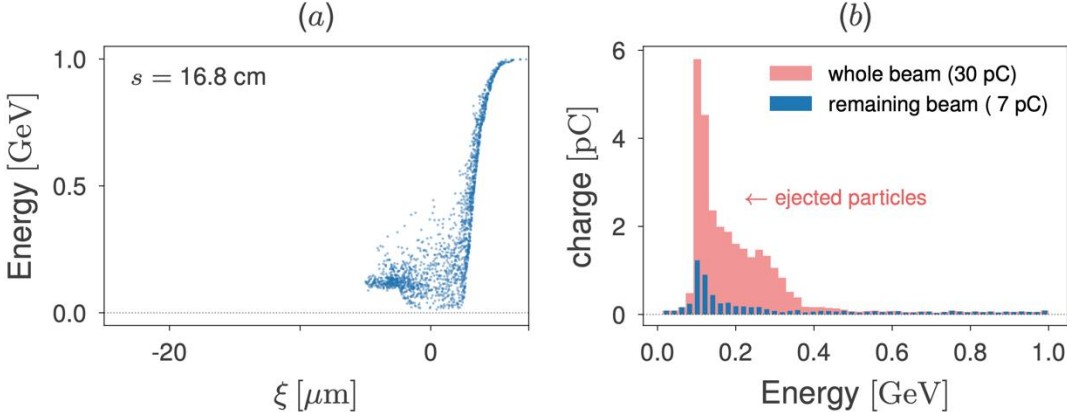

**Figure 3.** (**a**) The EuPRAXIA 1 GeV beam longitudinal phase space and (**b**) energy spectrum, after 16.8 cm propagation in plasma. At this propagation distance, particles with lower energies, which can be clearly seen in Figure 1a, are not present because they were ejected by the defocusing phase of the transverse wakefield. The energy spectrum for the remaining beam (7 pC, at this propagation distance) is represented by the blue columns in Figure 3b. In addition, the light-red shaded region shows the energy and charge distribution for the whole beam, including the transversely ejected particles.

Figure 4 gives the energy plots, as a function of propagation distance in plasma, for the density profile shown in Figure 2. The results show that the total beam energy reduces to less than 10% of its initial energy. Almost 80% of the initial beam energy is deposited in the plasma, and approximately 15% of the initial beam energy is transversely ejected. The average energy of the ejected particles is approximately 150 MeV. This tailored plasma density profile guarantees a relatively low beam energy deposited in the plasma vessel, ensuring safe operation of the plasma beam dump.

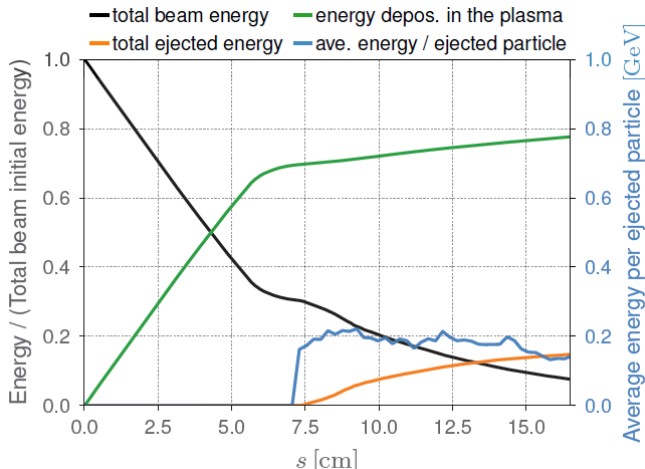

**Figure 4.** EuPRAXIA 1 GeV beam energy plots, as a function of propagation distance in plasma, for the density profile shown in Figure 2.

*3.2. Plasma Beam Dumps for the 5 GeV Beam*

When the electron beam energy reaches up to 5 GeV, the beam dump will be challenging if the conventional beam dump method is used, especially when the electron bunch is low emittance and ultrashort. For this reason, in this section, a passive plasma beam dump simulation is presented for the 5 GeV EuPRAXIA beam. For a highly relativistic beam, the rate of the total beam energy loss in uniform plasma is constant [12], only depending on the beam and plasma density profiles. In this way, since the beam and plasma parameters remain the same as in the previous case, the 5 GeV beam has to propagate for a longer distance in the plasma to reach the saturation distance, compared to the 1 GeV beam simulation.

Figure 5, in panel (a), shows the 5 GeV beam longitudinal phase space after 26 cm propagation in plasma with a density of $9.9 \times 10^{17}$ cm$^{-3}$, i.e., the same value adopted in the previous case. In panel (b), the corresponding energy spectrum for this propagation distance is also shown. Qualitatively, the phase space of Figure 5a is equivalent to that shown in Figure 1a for the 1 GeV beam; particles at the middle and tail of the bunch lose their energies, and some particles at the tail already started picking up energy from the wakefield. However, since the beam energy is 5-fold higher in this case, the propagation distance to reach this point is 26 cm, which is approximately 4.3-fold longer with the 6 cm observed in Figure 1a. Figure 5b demonstrates that the 5 GeV beam also shows a large number of particles decelerated to small energies. As in the 1 GeV beam, a smaller fraction of beam particles (the fraction within the bunch head vicinities in panel a) maintain their initial energies.

A plasma density tuning as shown in Figure 6 is adopted to mitigate the particle re-acceleration in the tail. In this case, the plasma density profile, which is constant up to $s = 26$ cm, is increased by a factor of 10 within a distance of 10 cm (from $s = 26$ cm to $s = 36$ cm). The effect of applying this tailored plasma density profile can be observed in Figure 7a, in which the longitudinal beam phase space is presented after 36 cm propagation in the plasma. Compared to the phase space at $s = 26$ cm, shown in Figure 5a, this figure shows that particles with lower energies at the beam tail were eliminated. In other words, the adoption of the plasma density profile from Figure 6 provides the same effect observed in the previous section for the 1 GeV beam. Figure 7b presents the energy spectrum associated with the beam phase space of Figure 7a. Apart from the distinct energy and charge scales, this beam energy spectrum is similar to that in Figure 1b.

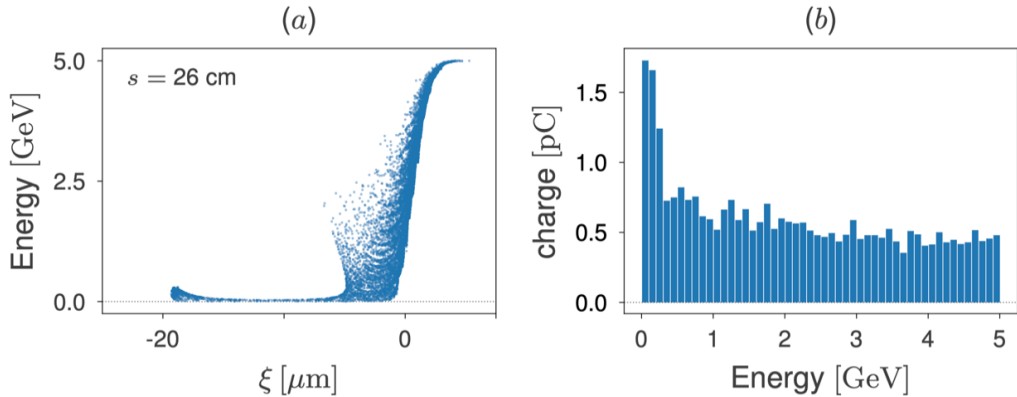

**Figure 5.** (**a**) The beam longitudinal phase space and (**b**) energy spectrum, after 26 cm propagation in plasma, for the EuPRAXIA 5 GeV beam.

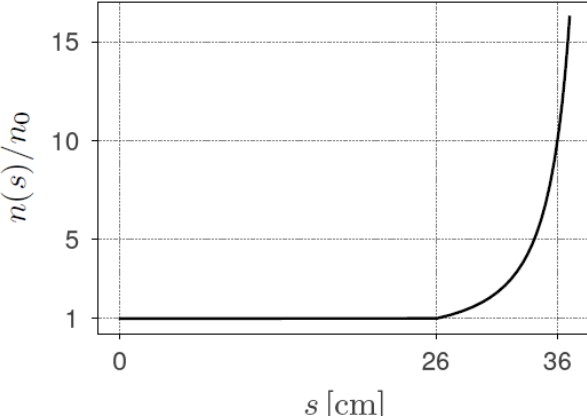

**Figure 6.** A tailored plasma density profile designed to eliminate the re-acceleration of particles in the bunch tail in the EuPRAXIA 5 GeV beam. The plasma density exhibits uniform behaviour until $s = 26$ cm, followed by a non-linear growth that reaches a 15-fold higher density at $s = 36$ cm, compared to the former uniform plasma density.

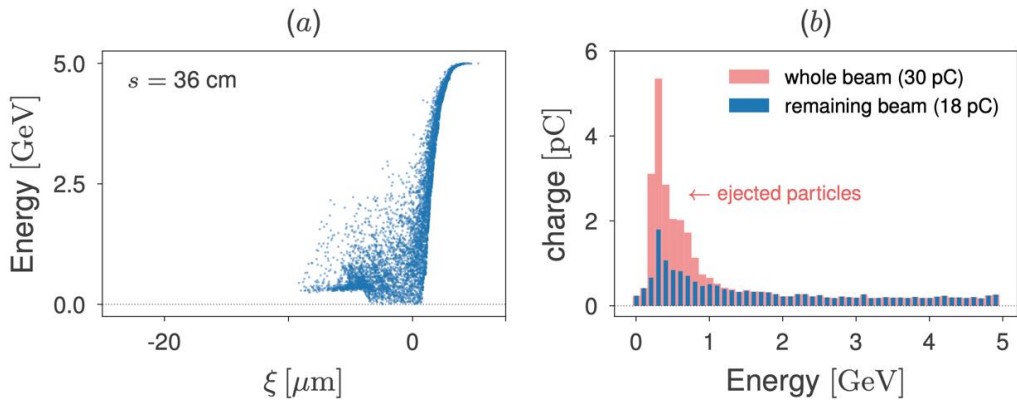

**Figure 7.** (**a**) The EuPRAXIA 5 GeV beam longitudinal phase space and (**b**) energy spectrum, after 36 cm propagation in plasma, for the EuPRAXIA 5 GeV beam. Compared to Figure 5a, this figure shows that the density profile from Figure 6 is effective in eliminating the lower energy particles reaching the accelerating wakefield phase, thus preventing the formation of a re-acceleration peak. The energy spectrum for the remaining beam (18 pC, at this propagation distance) is represented by the blue columns in panel (**b**). In addition, the light-red shaded region shows the energy and charge distribution for the whole beam, including the transversely ejected particles.

The beam energy loss as a function of the propagation distance in the plasma is shown in Figure 8. Clearly, it can be seen that, after a 37 cm propagation, the beam loses almost 80% of its initial energy—75% of the beam energy is deposited in the plasma, and 5% is carried out by the transversely ejected particles.

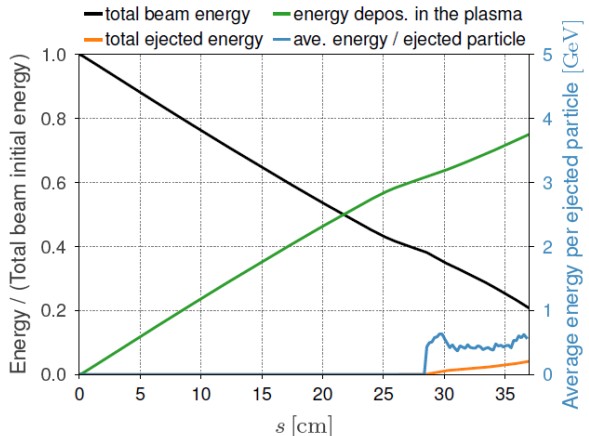

**Figure 8.** The energy plots as a function of propagation distance in plasma, for the plasma density profile shown in Figure 6 for the EuPRAXIA 5 GeV beam.

## 4. Discussion and Conclusions

The study shown here demonstrates the viability of the PPBD for the EuPRAXIA 1 and 5 GeV beams, respectively. This study shows that, for the 1 GeV beam, a PPBD that is 16.8 cm in length with a tailored plasma density profile can remove almost 90% of the total beam energy—80% absorbed by the plasma, and 10% is ejected with particles carrying average energies of ~150 MeV. On the other hand, for the 5 GeV beam, simulation results show that a plasma cell with a length of 37 cm can cope with 80% of the total beam energy—75% is deposited in the plasma, and 5% is transported by the ejected particles. Although the percentage of the total ejected energy is lower for the 5 GeV PPBD, compared to the 1 GeV beam, the average energy of the ejected particles must be considered. While 1 GeV PPBD particles are ejected with average energies of ~150 MeV, 5 GeV PPBD particles are ejected with average energies of ~500 MeV. When comparing both cases, the fraction of ejected particles in the 1 GeV beam (23 pC, corresponding to approximately 77% of the beam particles) was quite higher than the fraction observed in the 5 GeV beam (12 pC, which corresponds to 40% of the total beam charge). Small differences in both tailored plasma density profiles, as well as the propagation distances chosen for the plots of each case, might play a role in the difference observed. However, further investigation has to be performed in order to achieve a better understanding. Otherwise, it remains undoubted that, compared to conventional beam dumps, the adoption of the PPBD can help keep the overall facility compact and safer, as per the conceptual EuPRAXIA design precepts.

On the other hand, we have not discussed the APBD scheme here due to the complexity associated with its implementation. However, since the EuPRAXIA project requires laser infrastructure, an active beam dump might be a viable option. By using a laser-driven wakefield, in principle, almost 100% of the beam energy could be deposited in plasma. As for the next step, how to recycle or reuse the energy deposited in the plasma will be a key step forward. Interestingly, a recent experiment performed at Rutherford Appleton Laboratory (RAL) on a multiple laser pulse-driven plasma wakefield has shown possibility of energy recovery as a trailing laser pulse picking up energy from plasma [21].

The required technology for experimentally implementing plasma beam dumps, including a cooling system to address eventual plasma heating, is similar to the technology adopted to build plasma-based accelerators. A passive plasma beam dump could be naturally implemented in plasma wakefield accelerator (PWFA) or laser wakefield accelerator (LWFA) facilities, with the addition of a second plasma source (the plasma beam dump itself) with the desired plasma density profile.

On the other hand, the active plasma beam dump scheme is more suitable for LWFA facilities, since it requires laser infrastructure. The plasma source could be obtained by filling a sapphire capillary with the chosen gas (such as, for example, hydrogen) to be ionised. Ionisation can be achieved by using a laser pulse or electric discharges. The plasma density can be controlled by setting the gas pressure inside the capillary, and the delay between ionisation and beam arrival in the capillary. In a previous work [14], a brief discussion about the feasibility of implementing tailored plasma density profiles was introduced. Techniques such as heating the extremities of the plasma source at distinct temperatures, as carried out in the project AWAKE [22], to produce a 10 metre long tapered plasma density profile were mentioned. However, creating the proposed tailored plasma density profiles, with a 10-fold density increase along a few centimetres, would certainly be a challenging task at present. This is an active, evolving field of research, which is also relevant for plasma-based acceleration. Existing analytical expressions for the total beam energy loss in a plasma beam dump can be used to define the plasma length and density to be adopted to decelerate a given electron beam. The total beam energy loss could be measured with a spectrometer. By comparing the energy spectra of beams with and without the plasma, it should be possible to estimate the total beam energy loss when the plasma beam dump is operating.

Plasma heating also has to be properly addressed. For the passive plasma beam dump presented here, if 100% of the energy from the 5 GeV, 30 pC EuPRAXIA beam was deposited in the plasma, it would receive an average power of approximately 0.15 W, assuming a 1 Hz repetition rate. For the active case, in general, the heating is dominated by the energy dissipated by the laser pulse in the plasma [12]. The maximum repetition rate at which the plasma beam dump can operate will ultimately depend on the capability of the adopted cooling system to address resulting plasma heating. Finally, it is worth mentioning that the future development of techniques to recover energy from the plasma may turn the adoption of plasma beam dumps, in an important milestone, towards safer, greener, and compact facilities.

**Author Contributions:** Conceptualization, G.X., A.B. and R.P.N.; methodology, G.X., A.B. and R.P.N.; software, A.B., R.P.N., L.L., O.J., Y.Z., B.W., C.D. and X.W.; draft preparation, G.X.; writing—review and editing, all.; visualization, A.B., L.L. and O.J. All authors have read and agreed to the published version of the manuscript.

**Funding:** This research received European Union Horizon 2020 Research and Innovation Program under Grant Agreement No. 653782 EuPRAXIA.

**Acknowledgments:** This work was supported by the Cockcroft Institute Core Grant and the STFC. The authors acknowledge computing resources provided by STFC Scientific Computing Department's SCARF cluster, and the University of Manchester's CSF3 cluster. The author Alexandre Bonatto acknowledges the CNPq (Chamada Universal 427273/2016-1). Finally, the authors acknowledge the developers of FBPIC and OpenPMD for their contributions to the scientific community, as well as the support they provide to the users.

**Conflicts of Interest:** The authors declare no conflict of interest.

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
