# Peer review of "Plasma Beam Dumps for the EuPRAXIA Facility"

_instruments, doi:10.3390/instruments4020010_

Round 1

Reviewer 1 Report

The manuscript investigates a very interesting design, a plasma based beam dump for GeV level electron beams, by using PIC simulations. It explored some parameter regimes and could be useful for 1 GeV and 5 GeV electrons. This work is very interesting and will certainly be of interest to the community. However, there are a number of questions stays unclear, the authors should clarify them before the MS being considered to be published.

  • The authors should specify all the parameters in the simulations, for example, is it 2D simulation or 3D simulation? What is the simulation box size, spatial resolution? What is the boundary conditions?
  • From the comparison between figure 1 and 3, I do agree that electron beam loss energy during the propagation through the plasma. While, in the phase space, there is clearly high energy tail in the electron beam. How to deal with the remaining high energy tail? Could the authors comment on that?
  • Also, one few following question for figure 3, what is the percentage of low energy electrons, in terms of the numbers?
  • From the 1 GeV and 5 GeV simulations, the authors claims that 80% and 75% of the energy been absorbed by the plasmas. I wonder if the authors observed significant plasma heating? And if the heating has some impact on the plasma shape and in turn affect the final beam dump?
  • In figure 3 and 5, there is a similar step-like profile in the phase space low energy end, I wonder how does it form? Could the authors explain it?
  • One technical question, could the authors add the correspond colorbars for figure 1, 3, 5, and 7.

Author Response

Dear Reviewer 1,

Kind Regards,

Alexandre Bonatto

Reviewer 2 Report

Dear authors,

Please find attached review.

Respectfully,

Author Response

Dear Reviewer 2,

Kind Regards,

Alexandre Bonatto

Reviewer 3 Report

The submitted paper nicely presents the simulated performance of a plasma beam dump for the 1-5 GeV elcytron beam of the EuPRAXIA compact laser-plasma accelerator. This exercise progresses the recent work presented in ref.14 by the same authors, extending the simulation from 1 to 5 GeV.

The paper is well written and suitable structured, focused on the specific exercise, more with the style of a communication than of an extensive regular paper, and a result is that some steps are not clear enough as detailed below, but this is OK if some more detailed explanations could be added.

1) Line 77 “used in our studies are listed in Table.” – Please add table number

2) Table 1 “Angular divergence” – Please add the measurement unit: radian?

3) Line 84-86 “the wakefield excited is in the quasi-linear to nonlinear regime, and the whole beam is contained in the first phase of its self-driven wakefield” – This step is not clear enough for the reader not expert in the specific topic. Please expand explaining in some detail.

4) Line 93 “the beam energy at the head of bunch does not change, while the particles at the tail start to gain energy.” – Fig.1 deserves some more explanation: how is the complex structure between -10 and 0 um explained? How does this figure fit between fig. 1b and c in ref 14, produced in the same conditions, but with a significant difference at s=5.4cm compared to s=6cm? Finally, though clear enough, better indicating where the tail is.

5) Line 98 “In doing so, the defocusing phase of the transverse wakefield will move towards the low energy particles at the bunch tail” – same as comment 3: please expand explaining better

6) Line 101 & 118 “to 10 n0 = 9.9×1018 cm-3, at approximately 17 cm.” and “reaches a 10 times higher density at s = 16 cm,” – 17cm or 16cm ?.

7) line 104 “It is found that the re-acceleration peak (as shown in Figure 1) is eliminated, and particles continue to lose their energies in the plasma.” – is it possible to comment on the different distribution around 0 um?

8) line 107-108 – These numbers look different in fig.4: 12% -> less than 10%; 10% -> about 15%

9) In the last section few words could be spent on the feasibility of a tailored density profile, as suggested here.

Author Response

Dear Reviewer 3,

Kind Regards,

Alexandre Bonatto

Round 2

Reviewer 1 Report

I'd like to thank the authors for taking the time to address each of my concerns. Based on these changes, it is my opinion that this work, deserves to be published in Instruments.

However, there are few minor details that I would like to ask the authors to confirm in the final manuscript,

1) Figure 2, the author claim in line 160 as "...a 10 times higher density at s=16.8 cm", while I recall the value was 16 cm in the first version, could the authors double check the number?

2) Figure 3 (a), the energy axis is currently marked as 0 to 5 GeV, I thought this is a simulation for 1 GeV e beam, could the authors confirm it?

3) Line 214, "a factor of ~15...", if looking at the Figure 7 (a), it appears to be only a factor of ~10 from 26 to 36 cm

4) Line 258,  "...1 GeV beam (27 pC...", again, typo, should be 23 pc...

Author Response

Dear Reviewer 1:

Please see below our answers:

I'd like to thank the authors for taking the time to address each of my concerns. Based on these changes, it is my opinion that this work, deserves to be published in Instruments.

We are the ones who thank you for your time and attention to our work. Thank you for recommending our paper for publication in Instruments.

However, there are few minor details that I would like to ask the authors to confirm in the final manuscript,

Again, thank you for such a detailed review. Please accept my apologies for producing some typos in the first revised version.

1) Figure 2, the author claim in line 160 as "...a 10 times higher density at s=16.8 cm", while I recall the value was 16 cm in the first version, could the authors double check the number?

The correct value is 16.8 cm. In the original submission, Reviewer 3 pointed out that we mentioned it as "approximately 16 cm" and "approximately 17 cm", in two distinct points of the text. Therefore, we decided using 16.8 cm to address the point at which density has a 10-fold increase. 

2) Figure 3 (a), the energy axis is currently marked as 0 to 5 GeV, I thought this is a simulation for 1 GeV e beam, could the authors confirm it?

Indeed, the labels from the energy axis were wrong. Thank you for noticing this error.

3) Line 214, "a factor of ~15...", if looking at the Figure 7 (a), it appears to be only a factor of ~10 from 26 to 36 cm

You are correct (the 15x factor takes place at ~37 cm). The text has been corrected to "a factor of 10 ...".

4) Line 258,  "...1 GeV beam (27 pC...", again, typo, should be 23 pc...

Thank you: the value has been corrected to 23 pC.

Reviewer 3 Report

The revised version has significantly improved, as all comments by the referees have been suitably taken into account.

Author Response

The revised version has significantly improved, as all comments by the referees have been suitably taken into account.

Dear Reviewer 3,

Once more, thank you for helping us improving our work.

Alexandre Bonatto,

(On behalf of all authors)